# Coffee Berry Borer, *Hypothenemus hampei* (Ferrari) (Coleoptera: Curculionidae): Activity and Infestation in the High Mountain and Blue Mountain Regions of Jamaica

**DOI:** 10.3390/insects14080694

**Published:** 2023-08-05

**Authors:** Ameka Myrie, Tannice Hall, Denneko Luke, Bhaskar Rao Chinthapalli, Paula Tennant, Dwight Robinson

**Affiliations:** Department of Life Sciences, University of the West Indies, Mona, Kingston 7, Jamaica; myriemeka@gmail.com (A.M.); tannice.hall02@uwimona.edu.jm (T.H.); denneko.luke@uwimona.edu.jm (D.L.); bhaskar.chinthapalli@uwimona.edu.jm (B.R.C.); paula.tennant@uwimona.edu.jm (P.T.)

**Keywords:** bark beetle, borer activity, borer infestation, *Coffea arabica*, crop phenology, kairomone traps, weather variables

## Abstract

**Simple Summary:**

The coffee berry borer (CBB; *Hypothenemus hampei*) is an invasive beetle that causes extensive damage to coffee plantations worldwide. Controlling the CBB is difficult because it primarily resides inside coffee berries during its lifecycle, which limits the effectiveness of insecticide applications. Identifying periods of heightened female CBB flight activity can assist growers in making management decisions and evaluating integrated pest management programmes. This study monitored seasonal CBB activity using traps on coffee farms in the high mountain and Blue Mountain regions in Jamaica. Trap collection numbers were compared with berry infestation in the field. The highest CBB infestation levels occurred in November and October in the high mountain region and Blue Mountain region, respectively, coinciding with the presence of susceptible berries. CBB activity and infestation were similar in both study locations and were not significantly influenced by temperature or humidity; however, there was a notable correlation between CBB activity and infestation and the amount of rainfall. Differences in cultural control practices and cropping cycles were also observed between locations. The study lays the groundwork for understanding the dynamics of CBB populations in Jamaica, which is crucial for managing the beetle.

**Abstract:**

Jamaica produces coffee marketed as Blue Mountain and high mountain (grown outside the Blue Mountains). Since the discovery of the coffee berry borer (CBB; *Hypothenemus hampei*) in Jamaica in 1978, chemical control has traditionally been the primary approach used to protect the crop from the pest. However, in the last 20 years, there has been an effort to shift towards more sustainable management strategies. The study was conducted to determine CBB activity (trap catch) and field infestation on coffee farms in the high mountains and Blue Mountains of Jamaica, over a crop cycle. A total of 27,929 and 12,921 CBBs were captured at high mountain and Blue Mountain farms, respectively. Peak CBB activity occurred in April in the high mountain region (365 CBBs/trap/month) and February in the Blue Mountain region (129 CBBs/trap/month). The highest levels of infestation were in November (33%) and October (34%) in the high mountain region and Blue Mountain region, respectively. There was no significant difference in the patterns of CBB activity and infestation between the study locations, and neither were related to the temperature or relative humidity. However, there was a significant relationship with rainfall. These data suggest that the population dynamics of the CBB may involve complex interactions among weather conditions, berry development, and agronomic practices.

## 1. Introduction

Coffee (*Coffea arabica* L. Gentianales, Rubiaceae) production in the Caribbean has a long history dating back to the eighteenth century when it was introduced to the region [1,2,3,4]. The Caribbean’s warm, humid climate is ideal for growing the perennial crop species and countries such as Puerto Rico, the Dominican Republic, and Jamaica are renowned for producing top-quality coffee beans [5]. Jamaica’s Blue Mountain coffee, in particular, is one of the most sought-after and expensive coffees [6]. The crop is grown on steep sloping lands in the central mountainous regions of Jamaica [5]. Only coffee that is cultivated in the Blue Mountain area of Jamaica (St. Thomas, St. Andrew, and Portland) at altitudes of over 500 m above sea level (masl) to 2000 masl is licensed and sold as Blue Mountain coffee [7,8,9]. Coffee grown at lower elevations starting at 300 masl, outside of the Blue Mountain range, is sold as high mountain coffee [8]. In 2017, 8000 ha produced an estimated 6222 tons of beans valued at 0.1% of world production [10]. More than 80% of the smallholder farms produce coffee in Jamaica, most of which are family-run operations that rely on manual labour to harvest and implement management practices [11,12,13].

Even though it is a significant source of income, Jamaica’s coffee production has been steadily declining in recent years [14], particularly outside the Blue Mountains where coffee prices are much lower and farmers are primarily substituting coffee with other crops such as sugar cane, cocoa, and banana [7]. Other setbacks are due to socioeconomic problems such increased production costs [5] as well as increases in weather-related events [15] and diseases [16,17]. Diseases, mainly the coffee leaf rust (*Hemileia vastratix*) and insect pests, the coffee leaf miner (*Leucoptera coffeella*) [18], and the coffee berry borer (*Hypothenemus hampei*) [19,20], affect production.

The coffee berry borer (CBB) is widely recognised as the most destructive pest in numerous countries where coffee is grown [20,21,22,23]. The presence of the borer leads to substantial reductions in production, mainly due to decreases in berry weight, decreases in coffee yield due to the abscission of berries, and additional labour costs associated with separating out the lower-quality fruits [24,25,26,27,28,29,30]. Environmental conditions are linked to both borer distribution and crop damage. Studies indicate a correlation between the altitude at which shrubs are cultivated and the severity of borer infestation [31,32,33,34], between the number of borer individuals and temperature [33,35], and the number of borer individuals during dry conditions of cultivation compared to wet conditions [32,34,36,37]. Essentially, the metabolism of the CBB is influenced by prevailing conditions. High relative humidity (>90%), especially after rainfall, and increases in temperatures stimulate the emergence of the CBB female [19,38,39,40]. The development of the CBB requires temperatures within the range of 13.9–15 °C while the optimal temperature for CBB reproduction is 25–27 °C [33,41]. Additionally, the timing of the coffee crop development is affected by the microclimate of different altitudes [29]. As environmental factors vary across different areas and time periods, changes in phenology can affect beetle activity and infestation rates at different times of the year [42].

Historically, the methods used to manage the pest have not been environmentally sustainable leading to the emergence of issues of pesticide residues [43], among others. The current shift towards implementing more sustainable practices involves monitoring infestation levels and flight activity at different stages of the coffee season to help identify peak periods of pest activity. This information is crucial for farmers when making informed decisions about control strategies and implementing timely measures when the CBB is most susceptible [29,44]. CBB flight activity begins when mated female beetles leave their initial berries and infest new host berries [45]. Baited traps using methanol or ethanol, which mimic kairomones released by developing berries [46,47], are used to monitor CBB activity in various coffee-producing regions in Central America [48,49], Brazil [50,51], Hawaii [52], and Colombia [53]. Traps typically consist of a cylindrical container holding approximately one-third of its volume in water, along with a wetting agent. The attractant mixture, housed in a separate container positioned above the water, diffuses through openings located in the upper third of the trap. Beetles attracted to the trap enter through these openings, eventually falling into the water. Regular monitoring of the traps can help identify population spikes and anticipate potential outbreaks.

Unlike many of its neighbouring islands, Jamaica has faced a persistent CBB pest problem since the 1970s. The borer was detected in 1978 during the processing of beans that were meant for export [54]. Subsequent surveys between 1979 and 1982 reported its spread from a small area in the lowlands of central Jamaica to the entire coffee-growing region of the island [55]. Since these initial surveys, McCook [56] reported flight activity of the CBB in the high mountain and Blue Mountain regions and validated the effectiveness of using locally designed traps that utilised a blend of methanol and ethanol as a lure. However, to develop action thresholds for management based on adult trapping data, a correlation between trap catch data and field infestation levels is required. In this study, CBB trap catch and infestation data on coffee farms in the high mountains and Blue Mountains of Jamaica were determined. Weather data were collected to explore the impact of environmental parameters and characterise the heterogeneous conditions on the farms at the two locations.

## 2. Materials and Methods

### 2.1. Site Description

Data collection was conducted at Baron Hall Estate in the parish of Clarendon and Rosehill farm in St Andrew (Figure 1). Baron Hall cultivates high mountain *Coffea arabica* (95% Typica variety; otherwise, Catimor) in monoculture on 130 ha at an elevation of approximately 567 masl. Guango (*Samanea saman*) trees shade the coffee shrubs. Rosehill cultivates Blue Mountain *Coffea arabica* (mostly Typica, plus Caturra and Hybrid de Timor varieties), shaded by banana (*Musa* sp.) shrubs, on 35 ha at an elevation of 960 masl. Both study sites consisted of an amalgamation of sublet plots, which were separated by areas of natural vegetation, and accessible by farm roads. Selected sublets, spanning 1 hectare, were enclosed by natural vegetation, while the two rectangular sample plots measuring 0.1 ha were situated centrally within each sublet. The mature coffee shrubs within sample plots were planted 1.5 m apart within a row and 3 m between rows, resulting in an approximate density of 2000 shrubs per ha. Shrubs were cut back 4 years prior to the start of the study, but the rootstock was a minimum age of 8 years. No chemical control interventions were carried out at both sites during the study period. A random stratified method was used for the placement of eight traps and the identification of six shrubs in each plot (Figure 2), from which activity and infestation were determined over a 12-month period.

### 2.2. Data Collection

CBB activity was based on the number of individuals caught in 8 kairomone traps hung from sticks 1.6 to 1.8 m above ground (Figure 2). The traps were built from recycled 1.5 L polyethylene terephthalate water bottles, as designed by [56], and included 1:1 methanol–ethanol mixture as attractant. The attractant, in a plastic vial (with a 2 mm opening) attached to the handle of the trap, was 5 cm above the level of the soapy water. Earlier studies reported no distinction in the capture of these traps compared to the commercial ones. [56]. CBBs were collected from the traps fortnightly, then pooled to determine monthly activity. Traps were checked for quality, cleaned, and refilled with soapy water and attractant, before redeployment after each fortnightly CBB collection. Using a 10x hand lens, the contents of the traps were sorted and the CBBs counted. After it was established that 1 mL of CBB corresponded to 500 individuals, this benchmark was applied where the CBB counts from the traps exceeded 500.

CBB infestation was based on the presence of at least one CBB entry hole in berries at the CBB-susceptible stage (i.e., pimento berry size and later stages, Table 1, Figure 3). Six shrubs were selected, tagged, and assessed repeatedly over the study period. Berries present on 12 branches (located at different positions—upper, middle, and lower) of the selected shrubs were examined monthly and the percentage infestation determined [29,34,57,58,59]. The 12-month study started before the peak of the harvesting period and continued over the course of the crop cycle (Table 1). Dominant phenological stages (Figure 3) were recorded monthly during the study period (Table 1). The relationship of activity and infestation with rainfall, temperature, and humidity was explored. Hobo Data Loggers (H08-032-08) were used on each farm to record temperature and percentage humidity every 3 h. These data, collected monthly using a shuttle, were run on the ProBox software. Rainfall data from the weather stations closest to the farms were provided by the Meteorological Office of Jamaica.

### 2.3. Data Analysis

Statistical tests were carried out using R statistical language and environment [60]. A generalised additive mixed model (GAMM) using the mgcv package [61] was employed to evaluate the impact of environmental variables, i.e., rainfall, humidity, and temperature. An autoregressive moving average (ARMA) error structure was incorporated to address autocorrelation in the time series and heterogeneity of variance. The ARMA error structure was defined by two parameters: (i) the number of auto-regressive (AR) parameters (p), and (ii) the number of moving average (MA) parameters (q) [62]. The most suitable number of ARMA parameters based on AIC, AICc (AIC corrected for finite sample sizes), and BIC (Bayesian information criterion) values were first determined using the auto.arima function from the forecast package [63]. It was applied to the residuals from an ordinary/regular regression from a generalised additive model (GAM) that included the dependent variables (activity and infestation) and independent variables (environmental variables). After this, the ARMA values obtained from the auto.arima function were used in the GAMM. The GAMM was used to assess (1) the relationship between the three-month running mean time-series data for the independent and dependent variables, and (2) to model the trends in all-time-series data over time to determine their significance. In all cases, before accepting final models, the residuals (normalised for the GAMM) of the models were checked using the ACF and PACF to determine if they were white noise (random and not autocorrelated). Additionally, the 95% confidence intervals of the AR, MA, and model parameters (GAMM) were checked to ensure that they did not include zero. The latter indicated that the model contained an adequate number of explanatory variables, and there were no problems with the error structure.

## 3. Results

A total of 40,850 CBBs were collected from 16 traps that were evenly distributed across sites located at Baron Hall Estate (27,929 CBB) in the high mountains and Rosehill in the Blue Mountains (12,921 CBB). No other beetles were found in the traps. However, a small percentage (less than 1%) of ants, bees, grasshoppers, spiders, frogs, and lizards were observed.

One major flight event was observed at each location. The mean (±SE) activity in the high mountains of Baron Hall Estate steadily increased from October and peaked in April at 365 ± 55.9 CBBs/trap/month before decreasing two months later with <50 CBBs/trap/month. This peak activity at Baron Hall coincided with the appearance of pimento-sized berries (Figure 3). Conversely, at Rosehill, the highest CBB activity was observed between January and April, with February recording peak CBB activity at 129 ± 29.7 CBBs/trap/month. From May to December, activity levels were lower at <30 CBBs/trap/month. At Rosehill, peak activity coincided with the onset of flowering. There was no significant difference in the patterns of CBB activity and infestation between study locations.

The absence of infestation data between January and March at Baron Hall and February at Rosehill was due to the lack of CBB-susceptible berries (pimento-sized berries; Figure 3). Two peaks in infestation were observed during the study period. At Baron Hall Estate, infestation peaked in November (32 ± 11.3%) and June (18 ± 3.9%), while, at Rosehill coffee farm, peaks were observed in October (33 ± 13.1%) and July (27 ± 13.4%) (Figure 4). Both peaks in infestation at Baron Hall and Rosehill coincided with the presence of mature green and ripe berries.

Neither temperature nor relative humidity had a significant relationship with CBB activity or infestation at the two locations. Average temperatures ranged from 18 to 23 °C and 16 to 23 °C at Baron Hall and Rosehill, respectively. Relative humidity ranged from 60–80% at Baron Hall and 80 to 88% at Rosehill.

CBB activity and infestation were significantly related to rainfall (Figure 5, Table 2). The highest levels of rainfall were recorded in May (632 mm) and October (415 mm) at Baron Hall. The highest levels of rainfall were also seen at Rosehill during the same period (339 mm in May and 322 mm in October). Two peaks of infestation were recorded, with the highest infestation occurring after both the May and October periods of peak rainfall in Baron Hall (Figure 5). Conversely, only one rainfall peak coincided with infestation at Rosehill. An initial infestation in October occurred during the peak rainfall period, while a second infestation was recorded after peak rainfall in July.

At Baron Hall, the period of high CBB activity coincided with an increase in the prevalence of pinhead berries, peaking in April with the initial development of pimento-sized berries. This peak in CBB activity occurred prior to peak rainfall in May, while, at Rosehill, peak activity was observed at the onset of the primary flowering period, which is locally termed the dogteeth stage (Figure 2).

## 4. Discussion

The findings of the study indicate that the CBB population in the high mountain region (Baron Hall Estate) was greater than that found in the Blue Mountain region (Rosehill) by a difference of 46%. CBBs were active throughout the year at both locations, with infestation lagging behind peak activity. Infestation was not recorded during periods without susceptible berries. Temperature and relative humidity were unrelated to both CBB activity and infestation at the study sites; however, rainfall likely played an important role in CBB population dynamics (i.e., activity and infestation). The data suggest that CBB population dynamics may be a complex relationship among weather conditions, berry development, and agronomic practices.

In general, the estimated CBB activity based on trap numbers was lower than that recorded in other studies. The highest CBB/trap count observed in this study was 365. Counts of 1117 CBBs/trap were reported in Brazil [64]. There were greater than 250 CBBs/trap per week in Hawaii [30]. Consistent with other studies, a single major peak in CBB activity was observed during the crop cycle. This occurred during the dominant phenological stage of early fruit development in April at Baron Hall Estate. However, at Rosehill, this coincided with early flower development as well as the secondary harvest peak period. Rosehill has a prolonged harvest period as ripe berries may be present throughout the cropping cycle. In Brazil, where the unsynchronised ripening of berries also results in an increased number of harvests [65], trap catch was low in fields during fruit maturation between March and July, and increased in August when the CBB emerged from the dry berries that remained on the plants or were on the ground [64]. According to [30], the highest CBB trap catch was observed during the post-harvest period between late November and February. Johnson and Manoukis [66] describe a major peak in flight activity for well-managed and poorly managed farms in Hawaii from March to May, as well as a second smaller peak at the end of the season between late October and December.

In this study, infestation levels were highest during the months when the coffee shrubs had mature green or ripe berries. Presumably, the presence of residual berries on the ground also contributed to these levels. As reported by [67], residual berries on the ground could potentially serve as a source for CBBs. These findings on the CBB infestation levels were consistent with previous findings of 1% up to 35% in Mexico, Columbia, Brazil, and East Africa [58,64,68,69,70]. Higher levels (up to 95%) were reported from Puerto Rico where the CBB was recently introduced and pest management strategies were yet to be implemented [34,71]. On poorly managed farms in Hawaii, infestation levels averaged 63% across sampling dates and increased to 95% by the end of the harvest season [66].

Two peaks in infestation levels that occurred after peak CBB activity were identified at both locations and corresponded to the dominant phenological stages of mature green and ripe berries. This is inconsistent with observations in other regions like Hawaii, Brazil, and Columbia [30,53,64] where a single peak in infestation was reported. It is also noteworthy that the trap catch counts in these countries were significantly positively correlated with field infestation rates.

In this study, the first peak of infestation followed CBB activity, which likely led to the infestation of berries [30,51]. The increase in infestation may have triggered pest management interventions resulting in a reduction in infestation levels shortly thereafter. The second peak in infestation, which then occurred a few months later, likely corresponds with the emergence of new generations of CBBs which had not been impacted by the control measures that were previously applied [30,53,64,72]. Overall, high CBB infestation levels despite low activity or relative trap counts were observed in this study. In almost all cases in other regions, trap catch significantly positively correlates with field infestation rates [30,51,53,64,73]. The relatively low number of CBBs trapped in this study may have been influenced by variables associated with trapping and may not be an accurate measure of CBB activity. The effectiveness of traps is influenced by colour (red or white), the rate of elution and mixture of alcohols (ethanol: methanol; in a 3:1 or 1:1 ratio), location (0.5–1.5 m high), and other factors, including weather conditions [45,50,52,74]. However, previous studies in Jamaica [56] showed similar levels of efficacy between this trap design and commercial traps. Moreover, the numbers caught during the study were consistent with similar catches in traps being used in other coffee fields in the area for the purpose of monitoring CBB activity. In addition to trap efficiency, the lower numbers may be associated with the relatively small size of the plots separated by strips of natural vegetation. A combination of these factors, as well as the absence of chemical interventions and inadequate cultural control practices, may have led to high CBB infestation levels, despite the observed low activity or relative trap counts. Additionally, the prolonged harvest period and the inability to remove all the mature berries may have contributed to the persistence of ripe and residual berries throughout the cropping cycle, providing a favorable environment for CBB infestation. Aristizábal et al. [53] attributed the lower CBB flight activity and lower field infestation on small farms, which did not use insecticides, to a more efficient workforce that maintained low berry infestation rates of less than 5%, as well as cooler temperatures, lower planting densities, and the presence of banana plantings. Factors such as warmer and wetter conditions, higher planting densities, and the reduced efficiency of harvest workers were identified as contributing to higher levels of CBB infestation on larger farms where insecticides were used.

The significant relationship between rainfall and both activity and infestation levels was not surprising. Similar relationships have been obtained with studies conducted in other regions [30,38,64,66,72]. Cumulative rainfall was positively correlated with CBB flight, but flight seemed to be suppressed during periods of heavy rainfall. Female CBBs typically remain semi-inactive within older berries and emerge after the onset of the first rains in search of new berries to begin the next reproductive cycle [19]. In addition to recording significant relationships between rainfall and both activity and infestation levels, the studies mentioned earlier [30,38,64,66,72] also noted a significant relationship between CBB activity and infestation levels and temperature, as well as relative humidity. This was not found in the present study. Similar observations were recently recorded in the neighboring Puerto Rico. Ruiz-Diaz and Rodrigues [75] found a correlation between total CBB capture and rainfall at one of their two sites on the Agricultural Experiment Station of Adjuntas in Puerto Rico, which is situated at an altitude of 585 masl. They also recorded meteorological data such as relative humidity, temperature, and rainfall. They did not observe any significant differences between CBB capture and these environmental variables at the second site.

Notwithstanding, the effect of rainfall may not be direct, but, rather, secondary, due to its impact on crop phenology, which may also directly impact CBB populations. At the beginning of a new coffee crop, the presence of new berries with over 20% dry matter content are vulnerable to infestation by CBB from the previous season [45]. These CBBs originate from old berries on coffee shrubs (or on the ground). CBBs’ emergence and their subsequent infestation of susceptible berries (i.e., pimento-sized berries in the high mountains) are prompted by the start of the rains when old coffee berries become waterlogged and uninhabitable [19]. Earlier rainfall events would have initiated flowering. In the Blue Mountains, as in Brazil [65] and Colombia [76,77], conditions allow for continuous berry production (and two harvest seasons). This inevitably contributes to the reproduction of CBBs by providing a source of food throughout the crop cycle.

To our knowledge, this is the first report on CBB activity and infestation levels in Jamaica. The information is a precursor to understanding the dynamics of the CBB population, which is crucial for the development of economic thresholds. Economic thresholds will inform the timing of intervention strategies as part of integrated pest management programmes, ultimately helping to optimise the use of pest control methods and reduce the economic and environmental costs associated with pest management on coffee farms.

## Figures and Tables

**Figure 1 insects-14-00694-f001:**
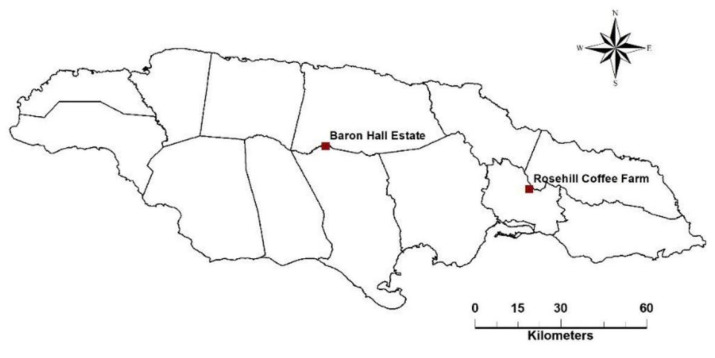
Locations of Baron Hall Estate (high mountain) and Rosehill (Blue Mountain) study sites.

**Figure 2 insects-14-00694-f002:**
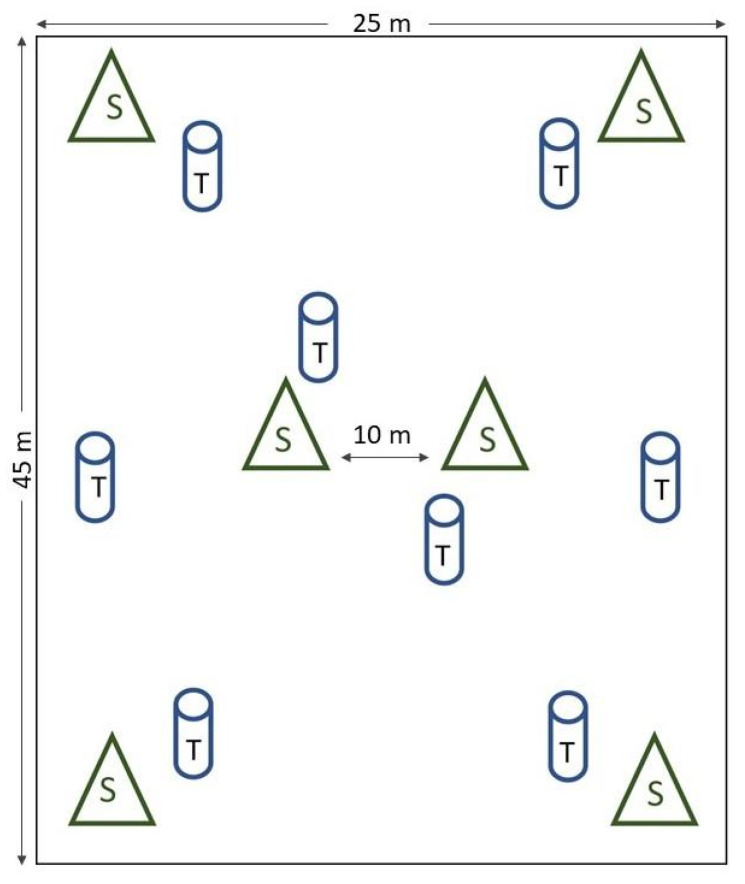
Placement of kairomone traps (T) and position of selected shrubs (S) for monitoring CBB activity and infestation levels, respectively, in 0.1 ha plots on Baron Hall Estate and Rosehill farms.

**Figure 3 insects-14-00694-f003:**
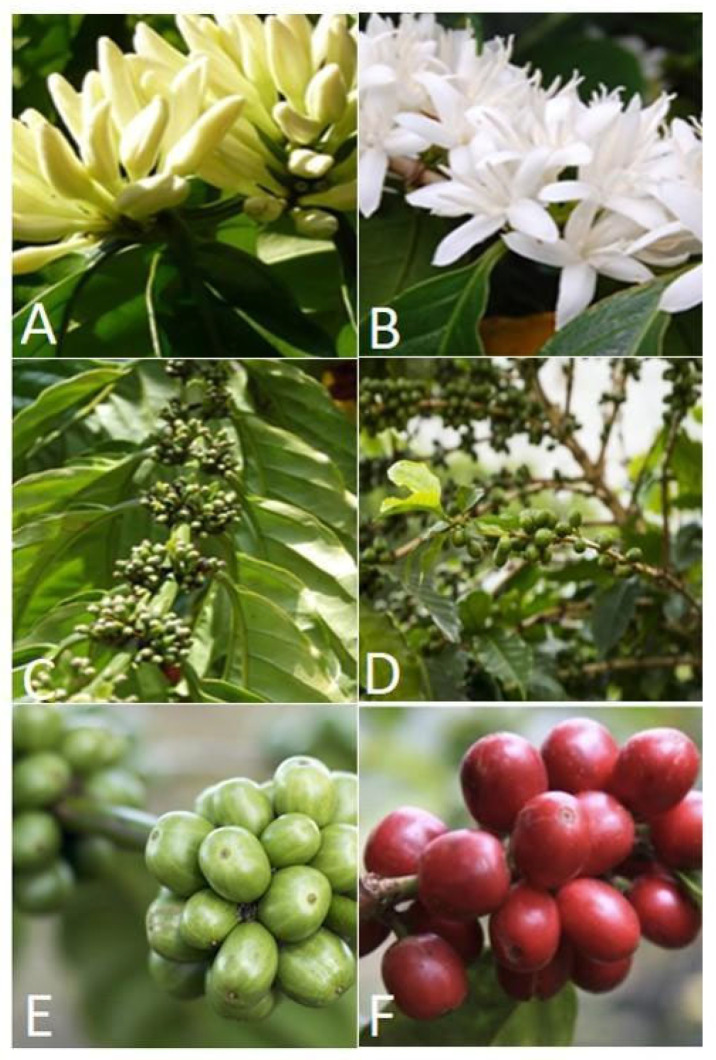
Phenological stages of the coffee crop as defined in Jamaica: (**A**) dogteeth, (**B**) flowering, (**C**) pinhead berries, (**D**) pimento-sized berries, (**E**) mature green berries, and (**F**) ripe berries (Jamaica Agricultural Commodities Regulatory Authority, pers. commun).

**Figure 4 insects-14-00694-f004:**
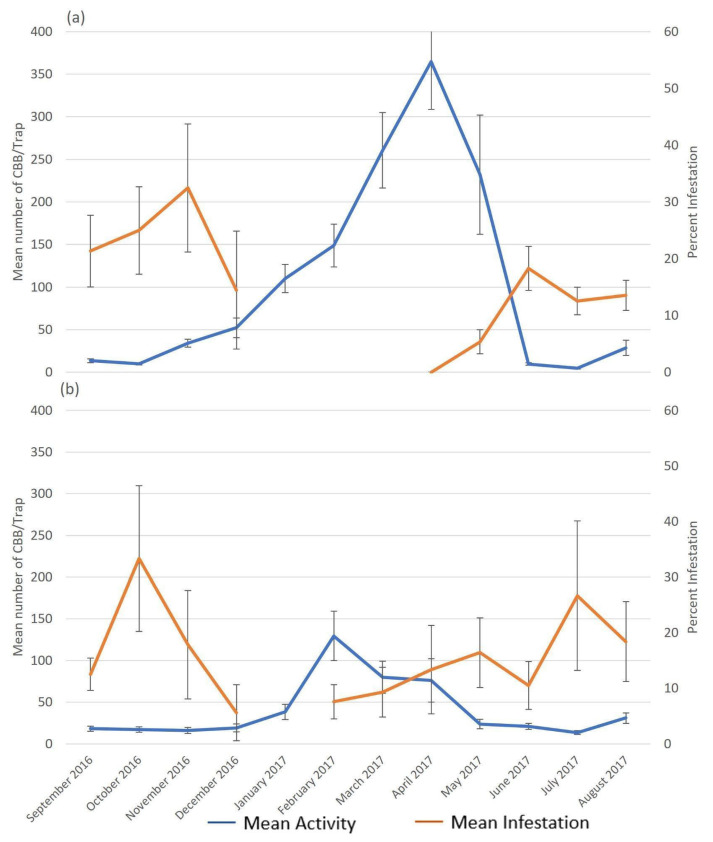
Mean (±SE) monthly CBB activity and infestation on (**a**) Baron Hall Estate in the high mountains and (**b**) Rosehill coffee farms in the Blue Mountains in Jamaica during the 2016–2017 crop cycle.

**Figure 5 insects-14-00694-f005:**
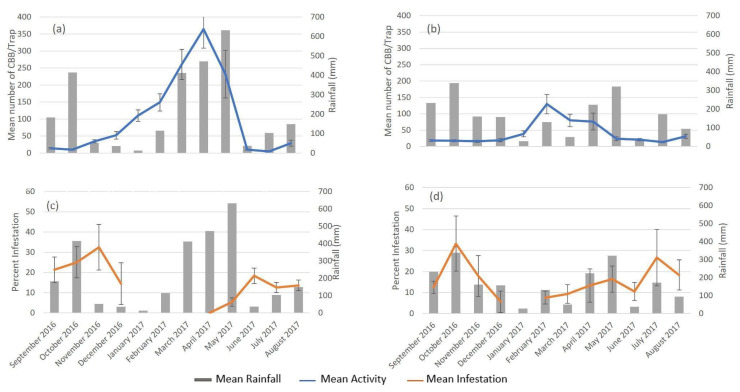
Mean (±SE) monthly rainfall with CBB (**a**) activity and (**c**) infestation on high mountain Baron Hall estate, and (**b**) activity and (**d**) infestation on Blue Mountain Rose Hill coffee farm during the 2016–2017 crop cycle.

**Table 1 insects-14-00694-t001:** Dominant crop phenology at time of data collection at Baron Hall Estate and Rosehill farms.

Date		Baron Hall Estate	Rosehill
2016	September	Mature green and ripe berries	Mature green and ripe berries
	October	Ripe berries	Ripe berries
	November	Ripe berries	Flowers and pinhead berries
	Dececember	Dogteeth	Pimento-sized berries
2017	January	Flowers	Dogteeth
	February	Flowers and pinhead berries	Dogteeth
	March	Pinhead berries	Flowers
	April	Pinhead and pimento-sized berries	Dogteeth and pinhead berries
	May	Pimento-sized berries	Pimento-sized berries
	Jun	Mature green berries	Flowers and mature green berries
	July	Mature green and ripe berries	Mature green berries
	August	Mature green and ripe berries	Mature green and ripe berries

**Table 2 insects-14-00694-t002:** Output from a generalised additive mixed model (GAMM), using three-month running mean data with an autoregressive moving average (ARMA) error structure to assess the influence of rainfall on CBB activity and infestation.

Correlation Structure and Parameters	Intercept	ARMA	F	P	Adjusted R2
Activity × Rainfall	73.03	-	4.648	<0.001	45.9%
Infestation × Rainfall	14.801	(1,0)	8.099	<0.001	40.1%

## Data Availability

The data presented in this study are available on request from the corresponding author.

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
