# Peer review of "Coffee Berry Borer, Hypothenemus hampei (Ferrari) (Coleoptera: Curculionidae): Activity and Infestation in the High Mountain and Blue Mountain Regions of Jamaica"

_insects, 2023, doi:10.3390/insects14080694_

Round 1

Reviewer 1 Report

 The authors' sampling effort should be acknowledged. Unfortunately, other data that allow new knowledge are required.

None

Author Response

We accepted the comments, reviewed the manuscript and made changes, that we think will address the general concern, in the resubmitted document.

Reviewer 2 Report

This MS represents the results of a study in field trapping and infestation in two coffee plots in Jamaica looking at a number of environmental conditions that might impact population dynamics and coffee beery infestation  The study plots occurred at two different elevations.

For the most part the study appears was well designed but the methodology for that study was not well documented in the study leading this reviewer to question some of the results and conclusion. with a revision of the methodology this MS will likely be acceptable for publication

Questions I have on the methodology are listed below and should be included in the MS revision.

1) Regarding the plots...where are the traps and plants used to measure infestation located relative to the whole plot (ie. middle of field at edge of plot etc?) what was the distance between each trap and each plant used to measure infestation? why were there only 8 traps and 6 plants in each plot study/ I would have liked to see more data points.

2) trapping.... how was the methanol/ethanol presented in the traps containing soapy water ?

3) Plants ....what percent of the coffee plants in the plot were sampled?   Was it the same plant sampled  throughout the study? how many berries were counted on each stem on the plant and were the infested berries removed or did you just count the ones with holes?? (I recommend a figure showing the location of the traps and plants relative to the whole plot at each location)

4) what was the status of any CBB control efforts at each plantation? did they use similar control methods and schedules and could that contribute to the results that you see in your analysis? (It would have been great if you could have compared this data to a farm that had no control or an abandoned farm to gauge the effects of any control measures)

Author Response

Reviewer 2

(1) Regarding the plots... where are the traps and plants used to measure infestation located relative to the whole plot (ie. middle of field at edge of plot etc?) what was the distance between each trap and each plant used to measure infestation? why were there only 8 traps and 6 plants in each plot study/ I would have liked to see more data points.

Both study sites consisted of an amalgamation of sublet plots, which were separated by areas of natural vegetation, and accessible through farm roads. Selected sublets, spanning 1 hectare, were enclosed by natural vegetation, while the sample plots measuring 0.1 ha were located centrally within each sublet. The modifications made to Figure 2 display the measurements accordingly. Trap and plant numbers were based on published recommendations.

Aristizabal, L. F., A. E. Bustillo, and S. P. Arthurs. 2016. Integrated pest management of coffee berry borer: strategies from Latin America that could be useful for coffee farmers in Hawaii. Insects. 7: 6.

Aristizábal, L.F.; Jiménez, M.; Bustillo, A.E.; Trujillo, H.I.; Arthurs, S.P. Monitoring Coffee Berry Borer, Hypothenemus hampei (Coleoptera: Curculionidae), Populations with Alcohol baited Funnel Traps in Coffee Farms in Colombia. Florida Entomol. 2015, 98, 381–83. doi:10.1653/024.098.0165.

Jaramillo, J., M. Setamou, E. Muchugu, A. Chabi-Olaye, A. Jaramillo, J. Mukabana, J. Maina, S. Gathara, and C. Borgemeister. 2013. Climate change or urbanization? Impacts on a traditional coffee production system in East Africa over the last 80 years. PLoS One. 8: e51815.

Larsen, A., and S. M. Philpott. 2010. Twig-nesting ants: the hidden predators of the coffee berry borer in Chiapas, Mexico. Biotropica 42: 342–347.

Mariño YA, Vega VJ, García JM, Verle Rodrigues JC, García NM, Bayman P. The Coffee Berry Borer (Coleoptera: Curculionidae) in Puerto Rico: Distribution, Infestation, and Population per Fruit. J Insect Sci. 2017 Jan 1;17(2):58. doi: 10.1093/jisesa/iew125. PMID: 28931153; PMCID: PMC5416771.

Pereira, A.E.; Vilela, E.F.; Tinoco, R.S.; de Lima, J.O.; Fantine, A.K.; Morais, E.G.; Franca, C.F.M. Correlation between numbers captured and infestation levels of the coffee berry-borer, Hypothenemus hampei: a preliminary basis for an action threshold using baited traps. Int. J. Pest Manage. 2012, 58, 183–190

(2) ... how was the methanol/ethanol presented in the traps containing soapy water?

A 15 mL vial containing the attractant was used in each trap. The vial was affixed to the handle of the trap and positioned 5 cm above the surface of the soapy water in the trap.

(3) Plants ....what percent of the coffee plants in the plot were sampled? Was it the same plant sampled throughout the study? how many berries were counted on each stem on the plant and were the infested berries removed or did you just count the ones with holes?? (I recommend a figure showing the location of the traps and plants relative to the whole plot at each location)

Coffee shrubs are planted at 1.5 m between plants within a row with 3 m between rows (approximate density of 2,000 shrubs per ha). Shrubs selected for sampling were tagged and monitored repeatedly over the study period. All berries on 12 branches (top, middle and bottom) were counted and examined for CBB entry holes.

(4) What was the status of any CBB control efforts at each plantation? did they use similar control methods and schedules and could that contribute to the results that you see in your analysis? (It would have been great if you could have compared this data to a farm that had no control or an abandoned farm to gauge the effects of any control measures)

Chemical control measures were not implemented during the study period at either of the sites.

Reviewer 3 Report

This manuscript is interesting and it support relevant information about CBB from Jamaica.  This manuscript needs a lot of improvements to be accepted for publication. 

Introduction:

Include a small paragraph about the use of alcohol-based traps. General characteristics and how they work. In what other coffee regions traps have been used for monitoring CBB? Why is it important to use alcohol-based traps to captures CBB? Why is important to understand the CBB flight activity and its relationship with coffee season?

Look at those references:

Dufour, B.P., and B. Frérot. 2008. Optimization of coffee berry borer, Hypothenemus hampei Ferrari (Coleoptera: Scolytidae), mass trapping with an attractant mixture. J. App. Entomol. 132: 591–600.

Cruz, R. E. N., and E. A. Malo. 2013. Chemical analysis of coffee berry volatiles that elicit an antennal response from the coffee berry borer Hypothenemus hampei. J. Mexican Chem. Soc. 57: 321–327.

Mendesil, E., T. J. A. Bruce, C. M. Woodcock, J. C. Caulfield, E. Seyoum, and J. A. Pickett. 2009. Semiochemicals used in host location by the coffee berry borer Hypothenemus hampei. J. Chem. Ecol. 35: 944–950.

Aristizábal L. F., Shriner S., Hollingsworth R. and Arthurs S. P. (2017) Flight activity and field infestation relationships for coffee berry borer Hypothenemus hampei (Ferrari) in commercial coffee plantations in Kona and Kau Districts, Hawaii. Journal of Economic Entomology 110, 2421–2427. doi: 10.1093/jee/tox215.

Fernandes F. L., Picanço M. C., Fernandes M. E. S., Dângelo R. A. C., Souza F. F. and Guedes R. N. C. (2015) A new and highly effective sampling plan using attractant-baited traps for the coffee berry borer (Hypothenemus hampei). Journal of Pest Science 88, 289–299.

Line 87. Site description: Density of trees per ha? Age of trees approximately? Agronomic activities that are normally conducted on those lots? Was any insecticide applied during this study? Be specific with that, yes or not, but it has to be included in the text. If the answer is yes, include the active ingredient, dose per Ha and type of sprayer used.

Line 104. CBB activity. How many traps were installed per coffee lot?, 8?, include the number of traps in this section.

Line 108. Evaluation of traps was conducted monthly? Correct? Or each two weeks? The cleaning and redeployment of traps was monthly?  So, be specific with intervals for CBB evaluation and cleaning.

Did you find other small beetles inside the traps? How you know the alcohol-based traps only capture CBB? Normally, other insects are attracted to those traps as well. You can give an estimation about other insects different to CBB that were found inside the traps.

Line 114. CBB infestation. Be most clear about how the evaluation was conducted. For example: number of trees that were selected for monitoring CBB (six trees? correct), include this number in this section.

What was the number of branches per tree that were counted? What specific berries were counted all berried or only developing green berries or ripe berries. For monitoring CBB infestation only green berries are counted; ripe berries are not counted since those ripe berries are expecting to be harvested.

For management of CBB any spray of insecticide is expecting to target CBB when females are colonizing and infesting berries.

Six trees for determine CBB infestation is a very low number of trees. However, if you have counted several branches per tree, the situation is better. So, this is why is important to be clear with the number of trees and branches counted.

Did you open infested berries to determine CBB penetration inside berries, known as CBB positions A, B, C, D?

Look at those references for CBB positions and monitoring CBB infestation:

Bustillo, A. E., M. R. Cardenas, D. Villalba, J. Orozco, M. P. Benavides, and F. J. Posada. 1998. Manejo integrado de la broca del café Hypothenemus hampei (Ferrari) en Colombia. Cenicafé, Chinchiná, Colombia. 134 pp.

Aristizábal, L.F.; Lara, O.; Arthurs, S.P. Implementing an integrated pest management for coffee berry borer in a specialty coffee plantation in Colombia. J. Integ. Pest Manag. 2012, 3, G1–G5

Jaramillo, J.; Borgemeister, C.; Baker, P. Coffee berry borer Hypothenemus hampei (Coleoptera: Curculionidae): Searching for sustainable control strategies. Bull. Entomol. Res. 2006, 96, 223–233.

Results:

Line 158, 161. 365 ± 55.9 CBB/trap per month?  129± 29.7 CBB/trap per month?

Line 168. How you explain the pick of CBB infestation (18 – 32%) in one farm and (27-33) in the other farms. Those CBB infestation are too high and the number of CBB captures is relative low.

Figures 5 & 6 are not needed. If in your data no relationship was find between temperature and relative humidity with CBB activity and CBB infestation, then those figures are not needed.

Discussion

This manuscript needs to include additional references to support its discussion.

How you can explain high CBB infestation levels and relative low number of CBB captured in traps.

I believe one aspect that cause this difference is related to the kind of alcohol used, specially the release of the alcohols per day. Methanol-ethanol enclose in transparent plastic bags, with a semipermeable material release a small amount of alcohol. So, CBB responses much better to very small amount of alcohols.

Look at this reference:

Aristizábal, L. F., M. Jiménez, A. E. Bustillo, H. I. Trujillo, and S. P. Arthurs. 2015. Monitoring coffee berry borer, Hypothenemus hampei (Coleoptera: Curculionidae), populations with alcohol-baited funnel traps in coffee farms in Colombia. Fla. Entomol. 98: 381–383.

In large coffee farms: CBB infestation (17-25%), traps captured 1,600 to 6,120 CBB per trap /week. In small coffee farms: CBB infestation (2-5%), traps captured 38-105 CBB per trap / week.

You have cited this reference, but look details in the information published.

Overall, I like this manuscript because there are very few reports or publication from Jamaica related with CBB and its management. This could be an interesting publication that open the doors to motivate additional studies about CBB in Jamaica, which I am totally sure are needed.

Author Response

Reviewer 3

(1) Introduction: Include a small paragraph about the use of alcohol-based traps. General characteristics and how they work. In what other coffee regions traps have been used for monitoring CBB? Why is it important to use alcohol-based traps to captures CBB? Why is important to understand the CBB flight activity and its relationship with coffee season?

Look at those references:

Dufour, B.P., and B. Frérot. 2008. Optimization of coffee berry borer, Hypothenemus hampei Ferrari (Coleoptera: Scolytidae), mass trapping with an attractant mixture. J. App. Entomol. 132: 591–600.

Cruz, R. E. N., and E. A. Malo. 2013. Chemical analysis of coffee berry volatiles that elicit an antennal response from the coffee berry borer Hypothenemus hampei. J. Mexican Chem. Soc. 57: 321–327.

Mendesil, E., T. J. A. Bruce, C. M. Woodcock, J. C. Caulfield, E. Seyoum, and J. A. Pickett. 2009. Semiochemicals used in host location by the coffee berry borer Hypothenemus hampei. J. Chem. Ecol. 35: 944–950.

Aristizábal L. F., Shriner S., Hollingsworth R. and Arthurs S. P. (2017) Flight activity and field infestation relationships for coffee berry borer Hypothenemus hampei (Ferrari) in commercial coffee plantations in Kona and Kau Districts, Hawaii. Journal of Economic Entomology 110, 2421–2427. doi: 10.1093/jee/tox215.

Fernandes F. L., Picanço M. C., Fernandes M. E. S., Dângelo R. A. C., Souza F. F. and Guedes R. N. C. (2015) A new and highly effective sampling plan using attractant-baited traps for the coffee berry borer (Hypothenemus hampei). Journal of Pest Science 88, 289–299.

The text below has been added to the Introduction.

Historically, methods used to manage the pest have not been environmentally sustainable leading to the emergence of issues of pesticide residues [Robinson and Mansingh 1999], among others. The current shift towards implementing more sustainable practices involves monitoring infestation levels and flight activity at different stages of the coffee season to help identify peak periods of pest activity. This information is crucial for farmers to make informed decisions about control strategies and implement timely measures when CBB is most susceptible [Aristizábal et al 2016, Kawabata et al 2017]. CBB flight activity begins when mated female beetles leave their initial berries and infest new host berries [Mathieu et al., 1997]. Baited traps using methanol or ethanol, which mimic kairomones released by developing berries [Mendesil et al. 2009, Cruz and Malo 2013) are used to monitor CBB activity in various coffee-producing regions in Central America [Dufour and Frérot 2008, Gutierrez-Martinez and Ondarza 1996], Brazil [Silva et al 2006, Fernandes et al 2011], Hawaii [Messing 2012, Aristizábal 2017] and Colombia [Aristizábal et al 2015]. Traps typically consist of a cylindrical container holding approximately one-third of its volume in water, along with a wetting agent. The attractant mixture, housed in a separate container positioned above the water, diffuses through openings located in the upper third of the trap. Beetles attracted to the trap enter through these openings, eventually falling into the water. Regular monitoring can help identify population spikes and anticipate potential outbreaks.

(2) Line 87. Site description: Density of trees per ha? Age of trees approximately? Agronomic activities that are normally conducted on those lots? Was any insecticide applied during this study? Be specific with that, yes or not, but it has to be included in the text. If the answer is yes, include the active ingredient, dose per Ha and type of sprayer used.

Coffee shrubs are planted at 1.5 m between plants within a row and 3 m between rows resulting in an approximate density of 2,000 shrubs per ha. Shrubs were cut back 4 years prior to the start of the study, however the rootstock is a minimum age of 8 years. No chemical control interventions were done during the study period at both sites.

(3) Line 104. CBB activity. How many traps were installed per coffee lot? 8?, include the number of traps in this section.

Eight traps were installed per coffee lot.

(4) Line 108. Evaluation of traps was conducted monthly? Correct? Or each two weeks? The cleaning and redeployment of traps was monthly? So, be specific with intervals for CBB evaluation and cleaning. Did you find other small beetles inside the traps? How you know the alcohol-based traps only capture CBB? Normally, other insects are attracted to those traps as well. You can give an estimation about other insects different to CBB that were found inside the traps.

Cleaning of traps and CBB evaluation were conducted every two weeks. No other beetles were found in the traps. A small percentage (less than 1%) of ants, bees, grasshoppers, spiders, frogs, and lizards were also observed in the traps.

(5) Line 114. CBB infestation. Be most clear about how the evaluation was conducted. For example: number of trees that were selected for monitoring CBB (six trees? correct), include this number in this section.

What was the number of branches per tree that were counted? What specific berries were counted all berried or only developing green berries or ripe berries. For monitoring CBB infestation only green berries are counted; ripe berries are not counted since those ripe berries are expecting to be harvested. For management of CBB any spray of insecticide is expecting to target CBB when females are colonizing and infesting berries. Six trees for determine CBB infestation is a very low number of trees. However, if you have counted several branches per tree, the situation is better. So, this is why is important to be clear with the number of trees and branches counted.

Six shrubs were selected, tagged and assessed repeatedly over the study period. Berries present on the selected shrubs (12 branches located at the upper, middle and lower) were examined monthly and the percentage infestation determined. CBB infestation was based on the presence of at least one CBB entry hole in berries at the CBB susceptible stage (i.e., pimento berry size and beyond, Table 1, Fig. 3). The developing green berry stage is referred to as the pimento size berry in Jamaica. While we accept that in other regions monitoring CBB infestation involves counting of only green berries, in Jamaica, ripe berries are included in the assessment because of the high variability and incompleteness of the harvesting process resulting in the conversion of these berries to dry residual berries with high levels of infestation.

(6) Did you open infested berries to determine CBB penetration inside berries, known as CBB positions A, B, C, D? Look at those references for CBB positions and monitoring CBB infestation:

Bustillo, A. E., M. R. Cardenas, D. Villalba, J. Orozco, M. P. Benavides, and F. J. Posada. 1998. Manejo integrado de la broca del café Hypothenemus hampei (Ferrari) en Colombia. Cenicafé, Chinchiná, Colombia. 134 pp.

Aristizábal, L.F.; Lara, O.; Arthurs, S.P. Implementing an integrated pest management for coffee berry borer in a specialty coffee plantation in Colombia. J. Integ. Pest Manag. 2012, 3, G1–G5

Jaramillo, J.; Borgemeister, C.; Baker, P. Coffee berry borer Hypothenemus hampei (Coleoptera: Curculionidae): Searching for sustainable control strategies. Bull. Entomol. Res. 2006, 96, 223–233.

In this particular study, once a berry was penetrated, regardless of CBB position, it was considered infested.

Results

(7) Line 158, 161. 365 ± 55.9 CBB/trap per month? 129± 29.7 CBB/trap per month?

The text has been adjusted to CBB/trap per month.

(8) Line 168. How you explain the pick of CBB infestation (18 – 32%) in one farm and (27-33) in the other farms. Those CBB infestation are too high and the number of CBB captures is relative low.

No chemical interventions were employed during the study period, and the cultural control practices implemented were deemed inadequate. This may have resulted in high CBB infestation levels despite low activity/relative trap counts.

(9) Figures 5 & 6 are not needed. If in your data no relationship was find between temperature and relative humidity with CBB activity and CBB infestation, then those figures are not needed.

Figures 5 and 6 have been removed.

Discussion

This manuscript needs to include additional references to support its discussion.

(10) How you can explain high CBB infestation levels and relative low number of CBB captured in traps. I believe one aspect that cause this difference is related to the kind of alcohol used, specially the release of the alcohols per day. Methanol-ethanol enclose in transparent plastic bags, with a semipermeable material release a small amount of alcohol. So, CBB responses much better to very small amount of alcohols.

Look at this reference:

Aristizábal, L. F., M. Jiménez, A. E. Bustillo, H. I. Trujillo, and S. P. Arthurs. 2015. Monitoring coffee berry borer, Hypothenemus hampei (Coleoptera: Curculionidae), populations with alcohol-baited funnel traps in coffee farms in Colombia. Fla. Entomol. 98: 381–383.

In large coffee farms: CBB infestation (17-25%), traps captured 1,600 to 6,120 CBB per trap /week. In small coffee farms: CBB infestation (2-5%), traps captured 38-105 CBB per trap / week. You have cited this reference, but look details in the information published.

Aristizábal et al. (2015) attributed lower CBB flight activity and lower field infestation on small farms, which did not use insecticides, to a more efficient workforce that maintained low berry infestation rates of less than 5%. The authors also considered contributing factors, including cooler temperatures, lower planting densities, and the association with banana crops, which might have played a role in reducing CBB populations. The authors suggested that factors such as warmer and wetter conditions, higher planting densities, and reduced efficiency of harvest workers might have played a role in higher CBB infestation levels on the larger farms where insecticides were used.

In this study, we focused on the patterns of change in activity and infestation levels over time rather than the absolute values. The traps were built from recycled 1.5 L polyethylene terephthalate water bottles, and included 1:1 methanol-ethanol mixture as attractant in a plastic 15 mL vial with a 2 mm diameter opening. The vial was attached to the handle of the trap and was 5 cm above the level of the soapy water. The 15 mL vial with a 2 mm diameter opening would allow for the release of small amounts of attractant was used in the traps. Previous studies in Jamaica (McCook 2007) showed similar levels of efficacy between this trap design and commercial traps. The effectiveness of traps is influenced by color (red or white), the elution rate and mixture of alcohols (ethanol: methanol; in a 3:1 or 1:1 ratio), and location (0.5–1.5 m high), and other factors, including weather conditions, and pest infestation levels as discussed in the studies listed below.

Messing R.H. The coffee berry borer (Hypothenemus hampei) invades Hawaii: Preliminary investigations on trap response alternate hosts. Insects. 2012;3:640–652. doi:10.3390/insects3030640.

Silva F.C.D., Ventura M.U., Mikami A.Y., da Silva F.C., Morales L. Capture of Hypothenemus hampei Ferrari (Coleoptera: Scolytidae) in response to trap characteristics. Sci. Agric. 2006;63:567–571. doi: 10.1590/S0103-90162006000600010.

Mathieu F., Brun L.O., Marcillaud C., Frérot B. Trapping of the coffee berry borer within a mesh-enclosed environment: Interaction of olfactory and visual stimuli. J. Appl. Entomol. 1997;121:181–186. doi: 10.1111/j.1439-0418.1997.tb01390.x.

Uemura-Lima D.H., Ventura M.U., Mikami A.Y., da Silva F.C., Morales L. Response of coffee berry borer, Hypothenemus hampei (Ferrari) (Coleoptera: Scolytidae) to vertical distributions of methanol: Ethanol traps. Neotrop. Entomol. 2010;39:930–933. doi: 10.1590/S1519-566X2010000600013.

We accept that the relatively low number of CBB trapped may be influenced by trap efficiency and may not be an accurate measure of CBB activity. However, the numbers caught during the study were consistent with similar catches in traps being used in other coffee fields in the area for the purpose of monitoring CBB activity. In addition to trap efficiency the lower numbers may be associated with the relatively small size of the plots separated by strips of natural vegetation. A combination of the factors mentioned above, as well as the absence of chemical interventions and inadequate cultural control practices, may have led to high CBB infestation levels in this study. This is despite the observed low activity or relative trap counts. Additionally, the prolonged harvest period and the inability to remove all the mature berries may have contributed to the persistence of ripe and residual berries throughout the cropping cycle, providing a favorable environment for CBB infestation.

The text has been adjusted accordingly.

Round 2

Reviewer 1 Report

My comments are in the text.

Author Response

  1. Simple Summary

This study monitored seasonal CBB activity using traps on coffee farms in high mountain and Blue Mountain regions in Jamaica. So what?

Since the initial detection of CBB in Jamaica in 1978, the primary method employed to safeguard the coffee crop from this pest has traditionally been chemical control. However, over the past two decades, a notable shift has occurred towards adopting more sustainable management strategies. In order to develop effective integrated pest management (IPM) measures for CBB, it is essential to have a comprehensive understanding of its activity patterns.

Aristizábal, L.F.; Bustillo, A.E.; Arthurs, S.P. Integrated pest management of coffee berry borer: strategies from Latin America that could be useful for coffee farmers in Hawaii. Insects. 2016, 7, 6, doi:10.3390/ insects7010006.

Kawabata, A.M.; Nakamoto, S.T.; Curtiss, R.T. Recommendations for Coffee Berry Borer Integrated Pest Management in Hawai’i 2016; Insect Pests IP-41; College of Tropical Agriculture and Human Resources, University of Hawai’i: Honolulu, HI, USA, 2017, p. 24.

  1. Abstract

The study was conducted to determine CBB activity (trap catch) and field infestation on coffee farms in high mountains and Blue Mountains of Jamaica, over a crop cycle. Coffee berry borer infestations change in time and space. What is the scientific contribution of this?

The study provides insights into CBB activity within the context of an island ecosystem. The findings indicate that there are common factors influencing both CBB infestation and coffee production in Jamaica, as well as in other coffee-producing regions globally. It represents the first of its kind conducted in Jamaica.

Even though [Jamaica is] not a major global producer [of coffee], the product’s competitive advantage is benchmarked by its high quality, the taste and appearance of the bean, contributing to it being ranked the number one brand on the international market. (Birthright 2016)

Birthwright, A.T. Liquid Gold or Poverty in a Cup? The Vulnerability of Blue Mountain and High Mountain Coffee Farmers in Jamaica to the Effects of Climate Change. 1st Edition: Climate Change and Food Security: Africa and the Caribbean. Thomas-Hope E., Ed.; Routledge Taylor and Francis Ltd. 2016, pp. 70-83)

  1. Introduction

Scattered and unfocused introduction

Considering the limited amount of published information available on coffee production in the Caribbean region, including Jamaica, we felt it was necessary to provide background information to establish the context for our study. And so, this section begins with a summary of coffee production locations on the island and the challenges encountered by coffee growers. Subsequently, we delved into the existing literature on the impact of CBB and the environmental factors influencing its distribution. We concluded the section by outlining the objectives of our study conducted in Jamaica.

While we acknowledge that the background information provided may not align with the conventional approach, we believe it plays a crucial role in establishing the context for the study.

  1. monitoring infestation levels and flight activity at different stages of the coffee season to help identify peak periods of pest activity. Rainfall plays an important ecological role on CBB populations. What is the new scientific knowledge? The population fluctuations of this pest are conditioned by rainfall, and the adults females leave the fruits only to infest the new crop. It is well known.

Indeed, this information is widely recognized. In this section of the Introduction, we have provided a summary of the existing knowledge on environmental factors that influence the distribution of CBB. We are, therefore, referencing established findings rather than presenting our own data in this particular context.

Reviewer 3 Report

This manuscript was improved by authors.  Only small details need to be addressed.  

See yellow highlight in the manuscript: corrections.

Abstract:  356 CBB/Trap / Month and 129 CBB/Trap/ Month.

Introduction:

You talk about elevation (masl) for do differentiation between the two locations (over 3000 m /Blue Mountain region and 1500-3000 m outside of Blue Mountain region.  I think it is correct, but in Material and Methods,  one farm is located at 567 masl (outside of Blue Mountain)  and the second (Rosehill/ Blue Mountain) at 960 masl.  The second one is supposedly to be > 3000 m correct? 

Good job!  

Author Response

This manuscript was improved by authors. Only small details need to be addressed. See yellow highlight in the manuscript: corrections.

  1. Abstract: 356 CBB/Trap/Month and 129 CBB/Trap/Month.

The text has been adjusted to CBB/trap/month.

  1. Introduction:

You talk about elevation (masl) for do differentiation between the two locations (over 3000 m /Blue Mountain region and 1500-3000 m outside of Blue Mountain region. I think it is correct, but in Material and Methods, one farm is located at 567 masl (outside of Blue Mountain) and the second (Rosehill/ Blue Mountain) at 960 masl. The second one is supposedly to be > 3000 m correct? Good job!

The numbers in the Introduction are not correct. Thank you for your eagle eye. The text has been revised to read as follows:

Only coffee that is cultivated in the Blue Mountain area of Jamaica (St. Thomas, St. Andrew and Portland) at altitudes of over 500 m above sea level (masl) to 2000 masl is licensed and sold as Blue Mountain coffee [7-9]. Coffee grown at lower elevations starting at 300 masl outside of the Blue Mountain range, is sold as high mountain [8]

We trust that the amendments made have satisfactorily addressed the concerns of the reviewers and look forward to your response. However, we did notice one item that requires clarification. On the webpage, it is mentioned that the email of coauthor, Tannice Hall, will not be published. We would appreciate it if you could explain the reason behind this statement. It is worth noting that the email provided for Dr. Hall is accurate.

Best regards,

Dwight Robinson